# Ensuring equitable access to quality HIV care for affected populations in complex sociocultural settings: Lessons from Nigeria

**Abdulsamad Salihu[1], Ibrahim Jahun[2]\*, David Olusegun Oyedeji[1], Wole Fajemisin[1]; Omokhudu Idogho[1], Samira Shehu[1], Jennifer Anyanti[1]**

**1** Society for Family Health, Abuja Nigeria, **2** Rady Faculty of Health Sciences, University of Manitoba, Winnipeg Canada.

\* drjahun@yahoo.co.uk; ibrahim.jahun@umanitoba.ca

## Abstract

### Background

HIV infection remains one of the major diseases of public health importance globally with an estimated 40.4 million deaths and 39 million people living with the virus by 2022. About 40 countries are on track to achieve a 95% reduction in AIDS-related mortality by 2030. This progress is however challenged by sub-optimal progress among affected populations (AP), also known as key populations (AP). Society for Family Health (SFH), with about 3 decades of experiences in AP program present in this paper an account of key strategies and innovations in adapting its service provisioning efforts to rapidly changing sociocultural and political barriers to service delivery among AP in northern Nigeria.

### Methods

SFH is an indigenous nonprofit, non-political, non-governmental organization in Nigeria that has pioneered HIV interventions among AP across most parts of Nigeria. SFH has successfully tailored its interventions to the unique cultural and religious diversity of Nigeria. The predominantly Islamic-orientated population in the northern part of the country and the Christian-oriented population in the southern part, which is culturally inclined to Western orientations, have all been considered in SFH's comprehensive approach instilling confidence in the effectiveness of its strategies. SFH implemented 3 key strategies to circumvent pervasive socio-cultural and political barriers that hindered successful AP program implementation in northern Nigeria by addressing structural barriers, systems barriers (service-provider and client-related barriers) and by deployment of innovations to optimize program performance. For the purposes of this retrospective cross-sectional study, deidentified routine aggregate program data was utilized to conduct secondary data analysis.

### Results

Between 2019 – 2023, SFH tested a total of 324,391 AP of whom 30,581 were found to be HIV positives yielding overall positivity rate of 9.4%. People who inject drugs (PWID) demonstrated sustained high positivity rate over the 5 years. About 80% of those initiated

**Data availability statement:** All data used in this study are made available withing the manuscript and its supporting documents (S1 Raw Data).

**Funding:** This study has been supported by the President's Emergency Plan for AIDS Relief (PEPFAR) through the United State Agency for International Development (USAID) under the terms of USAID Cooperative Agreement No. 7206201900009. The findings and conclusions in this report are those of the authors and do not necessarily represent the official position of the funding agencies.

**Competing interests:** The authors have declared that no competing interests exist.

on treatment were female sex workers (FSW) and men who have sex with men (MSM) contributing to 41.8% and 38.5% respectively. Year on year, the number of AP receiving ART more than doubled in 2020 and grew by 85%, 43% and 30% in 2021, 2022 and 2023 respectively. There was progressive increase in VL testing coverage between Year 1 – Year 3 across all the three AP typologies and then steady decline between Year 4 – Year 5. Between Year 1 – Year 2 the viral load suppression was at 91% with remarkable improvement to 97% in Year 3 and Year 4 and at 99% in Year 5.

## Conclusion

The implementation of people-centered, evidence-driven, culturally, and religiously sensitive program enabled SFH to reach a high number of AP in northern Nigeria. This helps improve equity in access to care by AP. There are specific program areas that need continuous improvement including strategies to reach MSM to avoid the evolution of new structural barriers; expansion of PWID programming to optimize all aspects of harm reduction; and sustained sensitization, education, and awareness creation among AP to improve uptake of PrEP and other prevention and care services.

## Introduction

The human immunodeficiency virus (HIV) infection remains one of the major diseases of public health importance globally with an estimated 40.4 million deaths and 39 million people (about 2.5 times the population of Lagos) living with the virus globally by the end of the year 2022 of which about two-thirds are in Africa [1]. Additionally, HIV remains a single disease entity with the highest funding in the history of global public health with about 140 billion US dollars spent from 2002 – 2022 [2]. The impact of this huge investment is enormous with an estimated 16.1 million averted deaths and about 27.5 million people living with HIV (PLHIV) on life-saving antiretroviral treatment (ART) at the end of 2021. Another important impact of the investment is that about 40 countries are on track to achieve a 95% reduction in AIDS-related mortality by 2030 of which nine of them are in eastern and southern Africa [3].

The progress towards global HIV control is however challenged by sub-optimal progress among some sub-populations such as affected populations (AP), and adolescent and young people (AYP). Affected populations (AP), also known as key populations (KP) are defined groups who due to specific higher-risk behaviors, are at an increased risk of HIV irrespective of the epidemic type or local context [4]. This group may include sex workers, gay men and other men who have sex with men (MSM), transgender people and people who inject drugs (PWID) [5]. The median HIV prevalence among AP is about 4 – 15 times that of general population and ranges from 2.5% (among sex workers) to 10.3% (among transgender persons) [5]. Globally, in 2021, AP and their sexual partners accounted for 70 percent of new HIV infections. One of the major factors contributing to sustained epidemiologic indicators among the AP is because they have inequitable access to safe, effective, and quality HIV services and face disproportionate levels of stigma, discrimination, violence, human rights violations, and criminalization [6].

With about two-thirds of the global HIV burden [1], Africa remains the epicenter of the HIV epidemic and therefore the collective effort of all stakeholders is essential to ensure sustained progress towards UNAIDS 95:95:95 targets. Nigeria ranks 4th globally after South Africa, Tanzania, and Mozambique in terms of HIV burden with an estimated 1.9 million PLHIV. The HIV prevalence among people aged 15 – 49 years is 1.4% [7], in line with the global trends,

the prevalence among AP in Nigeria is higher than the general population by over 10 folds: 15.5% among FSW, 25% among MSM, and 10.9% among PWID [8]. Therefore, attaining HIV epidemic control in Nigeria is unlikely without paying special attention to prevention, care, and treatment among the AP. Additionally, activities of these groups including all other LGBT related activities are criminalized in Nigeria, thereby posing serious challenges toward provision of effective HIV prevention and care among the AP subpopulation [9–11]. The Nigeria anti-gay law (same sex marriage prohibition Act 2014) carries a maximum sentence of 14 years in prison if convicted, bans same-sex marriage and other "amorous relationships," as well as LGBT organizations [12]. Enacting the law has not only complicated provision of quality HIV care to AP but has also perpetuated and exacerbated the longstanding pervasive stigma, discrimination, and violence to the subpopulation. Affected population infected with HIV are faced with compounding stigma based on being a AP and being HIV positive [13–15].

To effectively control HIV among AP, programs must be capable of circumventing these challenges. There is also a requirement to increase the number of organizations that provide AP friendly services to reach critical mass of service providers to meet the needs of AP across the breadth of Nigeria. Presently, there is a dearth of information about the availability and coverage of AP service providers that are adequately equipped and experienced [16] to address the barriers to care for AP that have been highlighted here. Lessons from the few well documented AP providers would be vital in catalyzing the growth of AP friendly service providers needed to fast-track epidemic control among this sub-population.

Society for Family Health (SFH), an indigenous Nigerian non-governmental organization with about 3 decades of experience in service provision among AP in Nigeria has been providing HIV care in the most complex terrains and circumstances in Nigeria. Therefore, the objective of this manuscript is to provide an account of SFH's evolution into a veritable AP service provider in Nigeria. This would include an account of key strategies and innovations in adapting its service provisioning efforts to rapidly changing socio-cultural and political barriers to service delivery among AP in different geopolitical regions of Nigeria. This paper will also provide insight into the gains recorded over these periods in terms of the size of AP reached with different HIV prevention and care services.

## Methods

### About Society for Family Health (SFH)

Is an indigenous nonprofit, non-political, non-governmental organization in Nigeria which has interventions in various health fields, including child survival, malaria prevention and treatment, HIV and AIDS prevention, and reproductive health. SFH has pioneered HIV interventions among AP across most parts of Nigeria. SFH's AP programs have addressed the full HIV response cascade of prevention, care and treatment, research, and policy while establishing community/patient-centered approaches to care and engagement. Since 1995, SFH has built a robust network of civil society organizations, community facilitators, and international NGOs who work with AP in Nigeria, demonstrating its inclusive and collaborative approach that makes everyone feel part of the solution. Having led AP interventions across more than 22 states in Nigeria over a period of about 3 decades, SFH has developed extensive knowledge and understanding of the local context of AP, providing solid foundation for its future interventions as described in subsequent sections of this manuscript.

### Settings and socio-cultural dynamics in Nigeria

Nigeria, a nation of extraordinary diversity, is home to over 250 distinct ethnic, linguistic, and religious groups, making it one of the most socially complex countries in Africa and is on track

to be the 4th most populous country by 2050 with projected population of 392 million people [17]. Notably, the Society for Family Health (SFH) has successfully tailored its interventions to the unique cultural and religious diversity of Nigeria. The predominantly Islamic-orientated population in the northern part of the country and the Christian-oriented population in the southern part, which is culturally inclined to Western orientations, have all been considered in SFH's comprehensive approach instilling confidence in the effectiveness of its strategies. This success paves the way for a more optimistic future in the fight against HIV and AIDS in Nigeria. Though AP activities are frowned at countrywide, people's perceptions and attitudes toward sexual minorities and other AP activities differ between these two geographical settings. Generally, sexual minorities find life more difficult in northern Nigeria than in the southern part of the country due to Sharia law adapted by predominantly Muslim states in the North [18]. Affected populations activities and programs are therefore necessarily designed keeping in mind these contexts to ensure effective implementation and long-term impact.

## Approach and information gathering

Primarily, SFH was an HIV prevention service provider but transitioned to comprehensive HIV service in 2014 ensuring a more holistic approach to care. In 2016, SFH initiated efforts to rollout AP-oriented contextual differentiated service delivery model (DSD) and identified the following barriers to effective HIV AP program implementation in Nigeria: structural barriers (criminalization of AP activities, stigma and discrimination), systems and client-related barriers such as long waiting and rigid working hours in health facilities, ineffective linkage of newly tested HIV positive AP to care, poor treatment monitoring due to large volume of patients receiving care, staff attitude and lack of understanding of AP program peculiarities as well as terrain and access to hard to reach areas. To address these barriers, SFH deployed a rigorous scientific approach to assess barriers' impact and to elicit best practices that could be adapted in designing strategies that would address them. Information was gathered using a mixed methods approach. Primary information emerged from a review of data collected through key in-depth informant interviews (KII), focus group discussions (FGD), surveys, project reports, website visits, and meetings with major HIV service providers, officials of State Agencies for the Control of AIDS and State AIDS Program Coordinators. Secondary data was gathered via desk review.

## Strategies

The strategies described in this manuscript were implemented in all states where SFH provides AP services, and they were tailored to suit the needs of all AP typologies. However, the data used for this study were obtained from six northern states (3 states each from northwest and northeast) as illustrated in Fig 1.

- **Addressing structural barriers:** HIV epidemics, particularly among AP, continue to be fueled by stigma and discrimination relating to homophobia, transphobia, sex work, drug use, gender inequality, violence, lack of community empowerment, inadequate understanding, and the interpretation of the laws by the AP themselves [16]. Other barriers include inadequate documentation of cases of violation of the human rights of AP by Law enforcement agencies (LEA), violations of human rights, and laws and policies criminalizing drug use sex work, and same-sex behavior [9–11].

- Several studies have shown that addressing structural barriers through multistakeholder engagement, education, and awareness creation among policymakers, civil society, and community-based organizations would improve access to care by AP [19–22]. Therefore, in

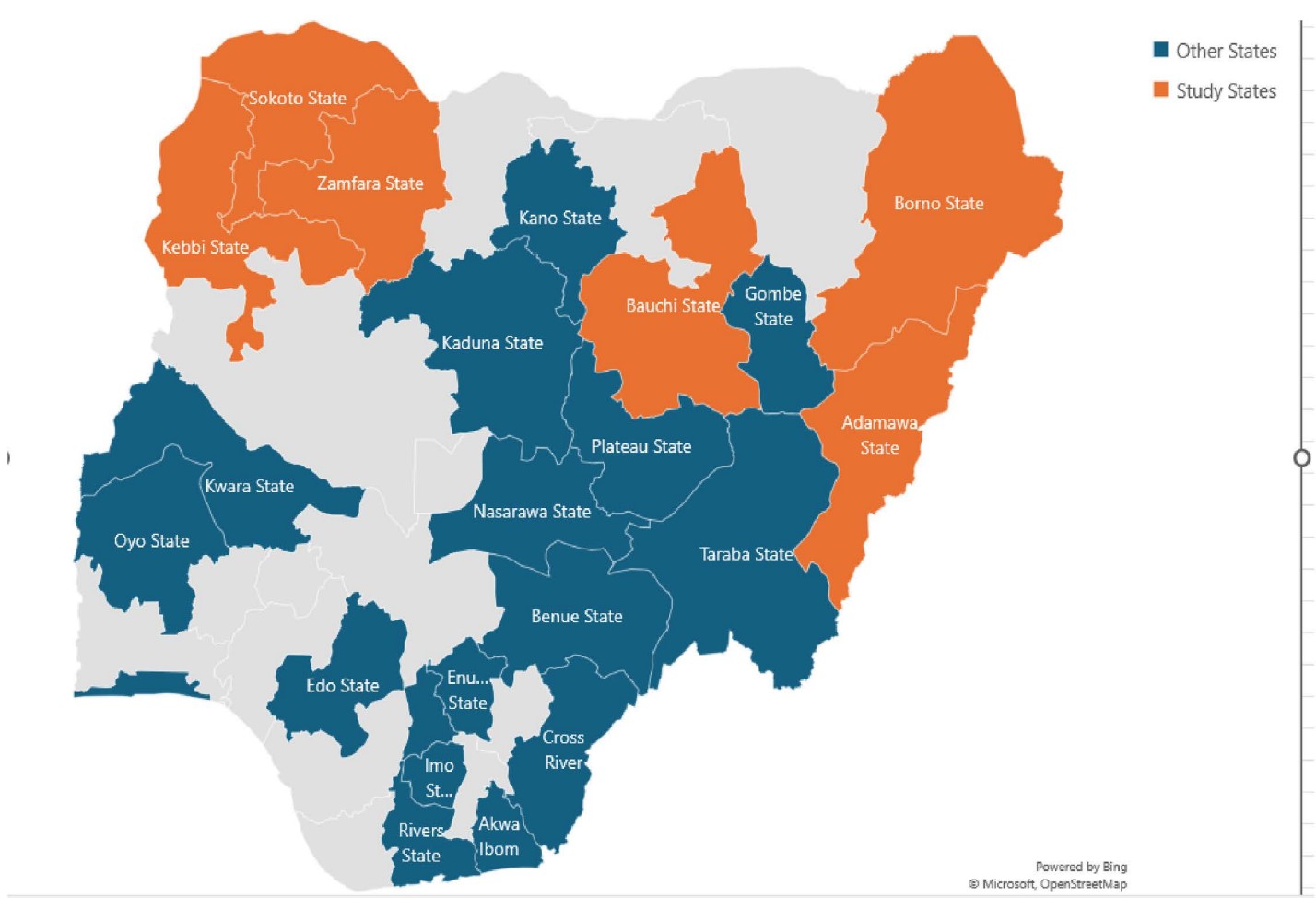

**Fig 1. Map of Nigeria Illustrating States where SFH Implemented Key Populations Program (Other States) and the Six Northern States (Study States).**

designing strategies to address the identified barriers, SFH uses a standardized human rights training curriculum, to enhance the legal literacy of the AP community. Also, in collaboration with relevant national and state HIV agencies, SFH supports improving policy and legal environments through tailored advocacy to relevant state organs and providing legal and paralegal services to AP community members. SFH also offers human rights interventions such as dedicated community funding for AP with the legal and human rights needs handled through legal bodies, increased funding to cover legal costs incurred while supporting AP with legal issues through strategic collaboration with the Legal Aid Council and other legal bodies on retainership basis – an arrangement in which SFH engages services of lawyers to serve affected AP on a recurring basis in exchange for a regular fee (retainer fee). SFH also documents the violation of the rights of the AP and uses these reports as a tool for advocating for stakeholders at different levels to inform program implementation. Fig 2 illustrates some key activities that are being implemented under these strategies.

- **Addressing systems barriers (service-provider and client-related barriers):** Service provider education and reorientation about AP needs, different models of DSD and client-centered approaches in HIV service provision promote access and service uptake by AP [20, 22–25]. SFH adapted these approaches and addressed the identified barriers by

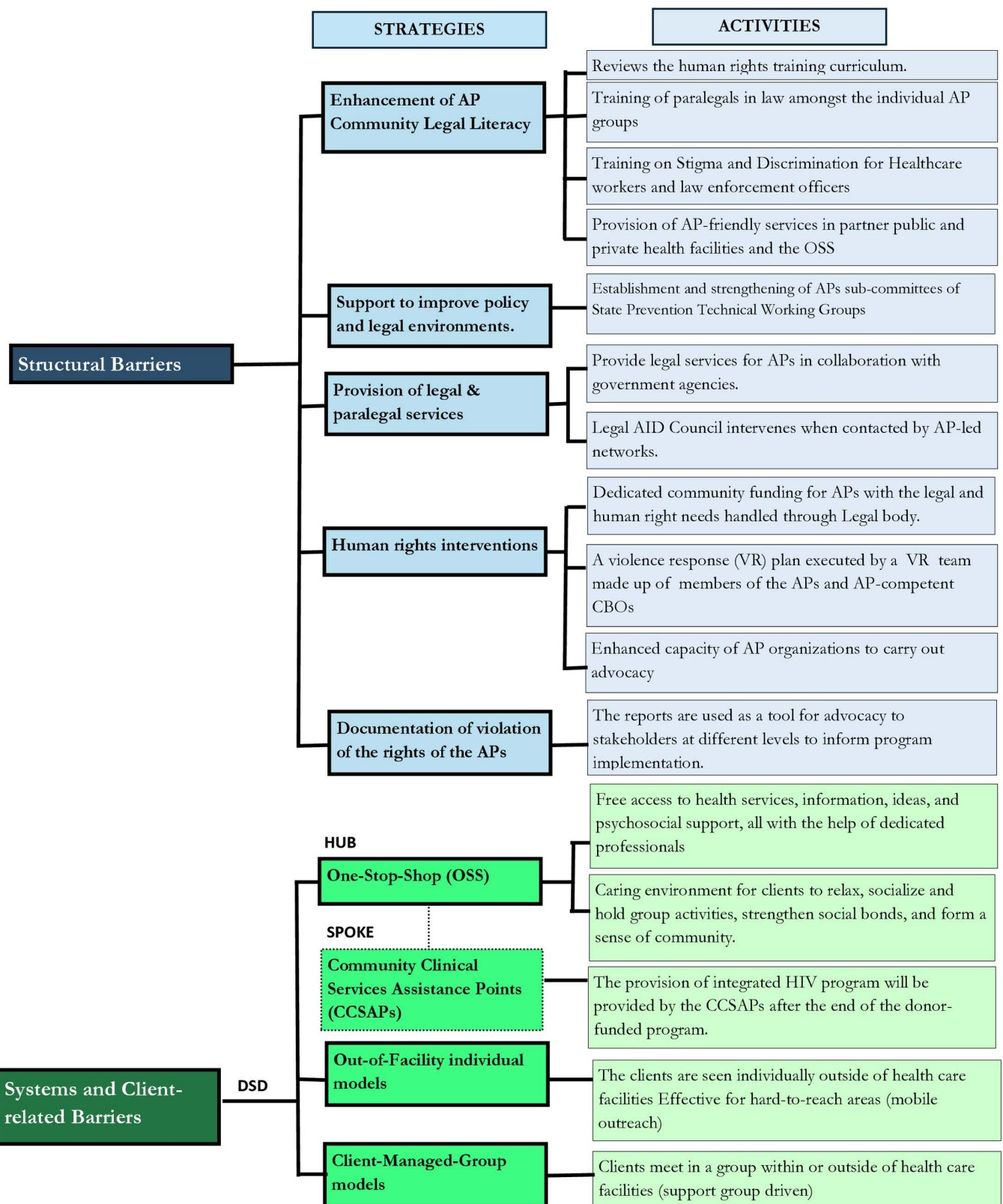

**Fig 2. Summary of SFH's Strategies in Addressing Structural and Systemic Barriers for Affected Populations Program Intervention in Nigeria.**

continuously reorienting facility staff on AP needs, legal and human rights, professional-ism and ethics which are periodically being conducted whenever deficiencies are observed among service providers. Additionally, to bring services closer to the AP communities, DSD was adopted as a community-based and decentralized care model. Society for Family Health entered a tripartite arrangement with state ministries of health, community-based organizations (CBO)/civil society organizations (CSO), and various ART health facilities to implement this strategy. Three categories of DSD models are being deployed: facility-based (one-stop-shop – OSS), out-of-facility individual and client-managed-group models.

The SFH OSS PLUS model is an open community space where AP can walk through the door and get free access to health services, information, ideas, and psychosocial support, all with the help of dedicated professionals. The SFH OSS team comprises of clinical supervisors who are medical doctors, nurses, laboratory scientists, pharmacists, counsellors, community nursing officers, data officers, receptionists, and office assistants. Additionally, there are case managers whose duty is to link positive clients to healthcare facilities and to track them when needed. A case manager manages the needs of a maximum of 100 clients providing person-alized care and support. The OSS also serves as a safe space and a client-focused facility that provides a welcoming and caring environment for clients to relax, socialize and hold group activities, strengthen social bonds, and form a sense of community, a platform for community mobilization, training and organizing activities. The OSS has proven to be an effective strategy in bringing services closer to AP and are well patronized by AP communities. However, running and maintaining OSS is expensive and may face sustainability challenges without donor-funding. Therefore, SFH introduces community clinical service assistance points (CCSAPs), which are AP-competent community-based public, private or faith-based facilities that provide rights-based and gender-responsive comprehensive HIV prevention, treatment, and care services to AP in a non-discriminatory manner. To ensure effective coordination of the CCSAPs and to strengthen their capacities, SFH made them serve as spokes to the OSS. The CCSAPs would also aid sustainability and ownership.

For the out-of-facility individual models, clients are seen individually outside of health care facilities to move care closer to clients in hard-to-reach locations. In this model, services are provided to clients directly by facility health workers (community nursing officers) at specific community locations operated either by community health workers or through a mobile out-reach service operated from the facility or OSS. For the 3rd DSD model, client-managed-group models, clients meet in a group within or outside of healthcare facilities. In SFH OSS facilities, in some cases, clients meet in groups, the healthcare workers are saddled with the responsibility of managing the provision of ART drug supply, care and support to groups of stable clients. Support group members meet monthly for 1-2 hour(s) in a session facilitated by health care workers with the case management officers. The Health Care Worker provides a brief symptom screen, referral where necessary, peer support and distribution of pre-packed ART to all the members present. Specific activities being implemented under the three DSD models are illustrated in Fig 2.

Interventions in any of the three DSD models are structured into three phases to address the specific needs of all categories of AP. The three phases are intensive, sustained and the transition or exit phase during which a client will transit to CCSAPs. The phase approach ensures that only stable and compliant clients are transitioned to CCSAPs as illustrated in Fig 3,

- **Program performance optimization:** In addition to the above strategies aimed at address-ing major barriers, SFH integrated innovations for optimal performance and attainment of annual program targets thereby reaching more AP. We instilled performance optimization

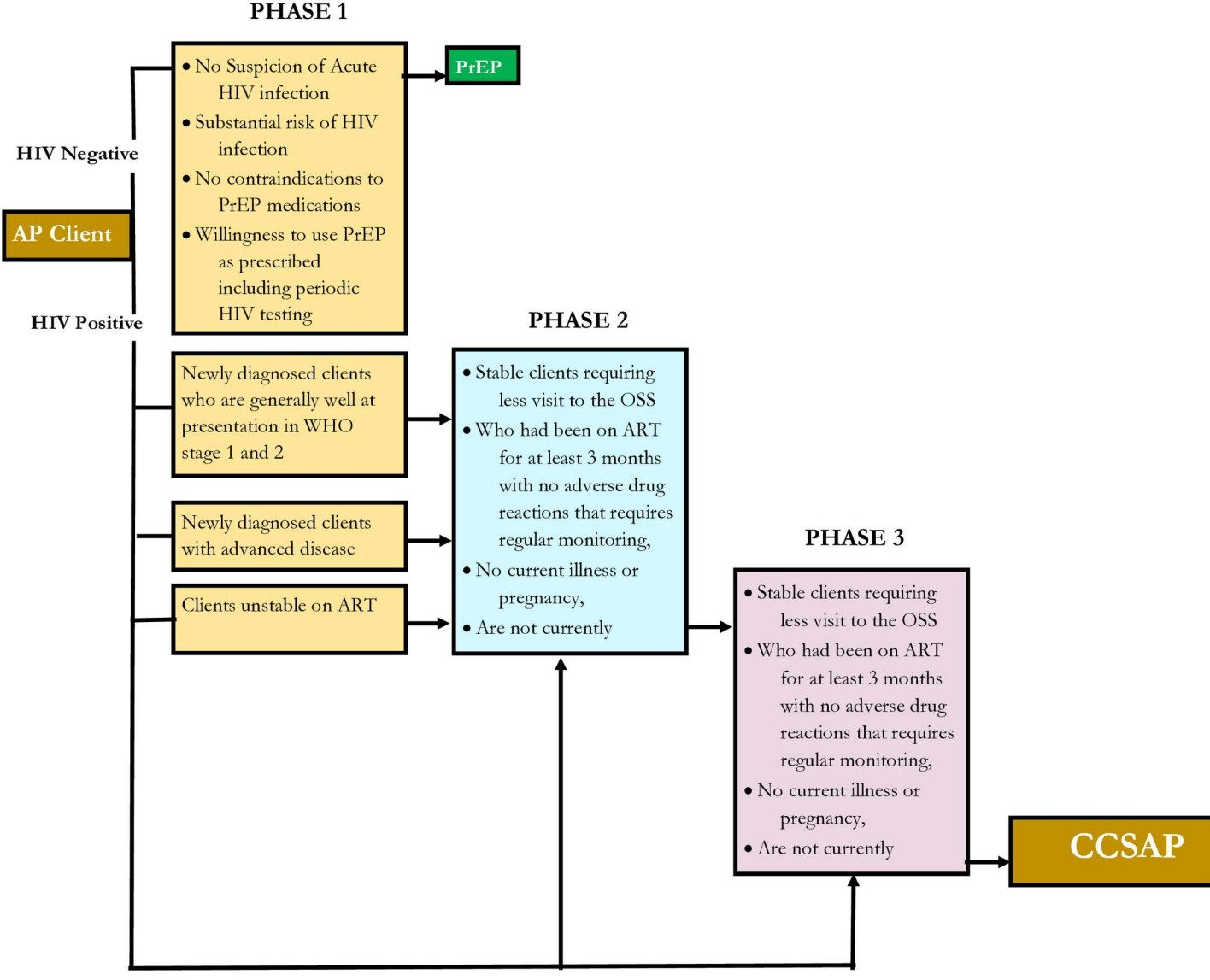

**Fig 3. The Three Phases of Differentiated Service Delivery (DSD) and Client Exit to CCSAP; SFH Affected Populations Service Delivery Model in Nigeria.**

strategies in each of the components of the three 95s of the UNAIDS 95-95-95 targets and pre-exposure prophylaxis (PrEP) through program surge.

For the 1st 95, SFH intensified case finding through index testing, sexual networks self-testing and ensured testing teams attend all AP social events. For the 2nd 95, we ensured strict compliance with the same day test and treat policy, and where contraindications existed or client declined, active linkage to care was employed to ensure treatment initiation at the appropriate time. We also optimized continuity of treatment by keeping in touch with newly identified clients for at least 8 weeks or until a client indicated full adherence to treatment requirements and the decentralized drug distribution model to ensure last mile distribution of essential ART to clients. Active tracking of defaulters and those interrupted in treatment (IIT) was intensified with enhanced adherence counselling and client education using the Undetectable=Untransmissible (U = U) message.

For the 3rd 95, major challenges affecting viral load (VL) monitoring and management such as failure to identify clients due for VL sample collection at the right time, clients not coming for sample collection at the right time, delayed return of VL results from PCR laboratories, were innovatively addressed. For example, SFH electronic medical record platform (LAMIS) is programmed to flag and display clients due for VL sample collection 4 weeks before the collection date and automatically send SMS reminders to the clients weekly. In addition, case managers follow up with clients to agree on collection dates. To minimize frequency of visits and save transportation costs for the clients, SFH ensures that VL collection date is aligned with ARV refills. Also, for clients who are unable to come for sample collection, we ensure reimbursement of transport fares. Remote sample login leveraging health information exchange technology for VL samples and return of results addressed the challenge of timely return of results from PCR laboratories. PrEP uptake optimization was through continuous clients' awareness, education and sensitization integrated in our DSD models and improved accountabilities for all clients tested, regular training of frontline staff on PrEP and introduction of PrEP champions.

In addition to the above performance optimization innovations, we integrated the following overarching initiatives to invigorate overall performance. Daily review meetings were held virtually with all case managers and peer navigators in each state, where best practices were shared, and challenges addressed thereby promoting cross-learning among frontline workers. Good performance is rewarded using non-monetary incentives such as T-shirts, hijabs, backpacks, and wristbands which generated healthy competition among case managers and peer navigators, thereby improving performance.

## Data management

Robust data management practices fundamentally drive SFH's successes. Quality data are collected for reporting purposes to national and funding agencies and for decision-making and program performance monitoring. Both paper and electronic data management processes are being used at different levels of service delivery. Effective data quality assurance mechanisms are well integrated for each of the two data management models to ensure data quality and integrity. Data used for decision-making through daily, weekly, and monthly meetings are conducted. Skilled data management staff are maintained through continuous training and mentorship. Continuous quality improvement initiatives are integrated into the system to strengthen efficiency and innovations.

Paper data management model until 2010, when the electronic reporting system was introduced in the HIV program in Nigeria, paper-based documentation and reporting were the predominant reporting model by the majority of healthcare facilities in line with the national reporting guidelines and policies [26] despite associated challenges such as data inconsistencies, data quality issues, missing records among others [27]. However, the national reporting guidelines and policies recommend maintaining paper-based documentation in parallel with electronic medical record (EMR), just like many other developed countries like Germany, Canada, and UK for legal, insurance and communication purposes, among others [28]. Therefore, in line with the national guidelines, the primary data collection at SFH's service delivery points (SDPs) is conducted using approved national data collection tools which include individual client management and monitoring forms, worksheets, registers, and summary forms. Demographic, clinical, and operational data are collected primarily using client-level forms and collated into appropriate registers.

Electronic data management Real-time data monitoring for client management through the continuum of care, warehousing of client-level data in data warehouse or data repository

for state, regional or national level analysis to inform HIV surveillance, evaluations, and epidemic control monitoring can only be achieved through deployment of effective EMR [29]. Between 2010 and 2012, the U.S. President's Emergency Plan for AIDS Relief (PEPFAR) supported the rollout of healthcare facility EMR systems [26] in Nigeria and by mid-2021 about 1,985 healthcare facilities have EMR deployed [29]. Two major EMR platforms are being used in HIV program in Nigeria – a locally developed Lafiya Management Information System (LAMIS) [30] and NigeriaMRS (NMRS), a customized OpenMRS platform [31]. SFH uses LAMIS, allowing data management assistants to enter paper-based client records manually. LAMIS supports both point of care (POC) services and retrospective data entry. The platform enables SFH to track clients across the continuum of care, generate data for improving clinical care informing DSD approaches and enabling cohort analyses and program monitoring.

Data use for decision making Evidence-based and data-driven programs, when implemented through rigorous, consistent and periodic data review meetings have proven to yield desired results [32–34]. SFH conducts daily data review meetings at SDPs across all OSS and CCSAPs to discuss challenges and best practices. Operational indicators such as linkage-to-care are closely monitored to ensure all clients identified through mobile outreaches are successfully linked and initiated on ART where applicable. At facility-level (OSS and CCSAPs) similar meetings are conducted on a weekly basis and in addition, performance progress toward targets is reviewed and targets recalibrated. Similarly, these meetings are conducted at state-level monthly, and quarterly at national level.

Study design and data access: This was retrospective cross-sectional study using deidentified aggregate routine program data from Jan 2019 – December 2023. The data was accessed during the study period from April – June 2024. As the data is deidentified, the individual records and information of the participants could not be linked or traced back by either the authors or any other person who accessed the data. Additionally, the data had earlier been made publicly available in routine annual reports by both the Federal Ministry of Health and the funder.

Data collection and analysis: Custom reports were generated from electronic medical records platform (LAMIS), using queries that included variables of interest over 5 years (Jan 2019 – December 2023). Variables of interests covered 95:95:95 cascade and PrEP as follows:

- Number of AP tested HIV positive: this represents all AP that were tested positive for HIV using standard national testing algorithm between January 2019 – December 2023, disaggregated by AP typology.

- Number of AP linked to care: of the AP who tested HIV positive, those that were successfully enrolled into care in approved HIV care facilities (including AP care facilities).

- Number of AP started on ART: of the AP successfully linked to care in approved healthcare facilities (including AP care facilities), those that were initiated on ART.

- Number of AP receiving ART: the number of AP actively on ART. It excludes AP started on ART and were documented as transferred out, died, stopped treatment or lost to follow-up.

- Number of AP whose VL sample were collected and tested: of the AP who are active on ART, those successfully bled and tested for VL at the appropriate time in line with national guidelines.

- Number of AP who are virally suppressed: this represents the number of AP on treatment who have documented VL results in their records from an approved PCR laboratory and the results in line with national guidelines were regarded as virally suppressed.

- Number of AP enrolled on PrEP: this represents the number of AP who were successfully enrolled on PrEP disaggregated by AP typology.

Data was retrieved and analyzed between April – July 2024. Aggregated variables were reviewed and validated for consistency using monthly historical reports (paper-based) that were reported previously to government and donors. In a situation where discrepancies were observed, LAMIS custom queries were reviewed, errors corrected, and custom reports regenerated. After validation, using Microsoft Excel, the following rates were calculated:

- Positivity rate: this is the proportion (expressed as %) of the number of AP who tested positive for HIV out of the total number of AP tested for HIV using standard national guidelines and testing algorithms during the study period.

- Linkage rate: this is the proportion (expressed as %) of the number of AP who were successfully enrolled in care in approved HIV care facilities (including AP care facilities) out of the total number of AP who were tested positive for HIV during the study period.

- VL testing coverage: this is the proportion (expressed as %) of AP whose samples were collected and tested out of the total number of AP who were eligible for VL test at a given time in line with national VL testing guidelines.

- VL suppression rate: this is the proportion (expressed as %) of the number of AP with suppressed VL test results out of the total number of AP whose samples were collected and tested during the study period.

## Results

Over a period of 5 years between 2019 – 2023, SFH tested a total of 324,391 AP of whom 30,581 were found to be HIV positives, range: 4,599 (2019) – 8,474 (2023), yielding overall positivity rate of 9.4%. as shown in Table 1.

The positivity rate ranges between 6.8% (Year 3) to 17.0% (Year 4). PWID demonstrated sustained high positivity rate over the 5 years ranging from 7.1% (Year 3) to 20.9% (Year 2). Within the same period, up to 30,772 AP initiated ART with an overall linkage rate of 100.6%. The least linkage rate recorded was 96.0% (Year 1) after which the linkage rate progressively increased to 99.7% in Year 3 and 100.7% in Year 5, a spike of 110.6% was observed in Year 2, as illustrated in Fig 4.

About 80% of the AP who initiated treatment were FSW and MSM contributing to 41.8% and 38.5% respectively. The total number of AP receiving ART (current on ART) demonstrated steady growth over the 5-year period. Year on year, the number of AP receiving ART more than doubled in 2020 and grew by 85%, 43% and 30% in 2021, 2022 and 2023 respectively as illustrated in Fig 5. There was progressive increase in VL testing coverage between Year 1 – Year 3 across all the three AP typologies and then steady decline between Year 4 – Year 5. Viral suppression is sustained at above 90% throughout the 5-year period. Between Year 1 – Year 2 the VLS was at 91% with remarkable improvement to 97% in Year 3 and Year 4 and at 99% in Year 5 (Table 1).

Rollout of pre-exposure prophylaxis (PrEP) remained very low in the first 2 years with breakthrough in the 3rd year whereby the uptake was 8.7 times higher than Year 2, 750 versus 6,544 respectively. Similarly, there was exponential growth in uptake between Year 4 and Year 5. PWID demonstrated a higher uptake consistently throughout the 5-year period as shown in Fig 6.

## Discussion

Deployment of SFH's combined strategies has resulted in identification of over 30,000 AP in the supported states in Northern Nigeria despite all the stringent socio-cultural barriers. AP activities are widely frowned upon in Nigeria with intensity of the aversion being highest in

**Table 1. Cascade of Care among Affected Populations (MSM, FSW, PWID) Receiving HIV Care in SFH Supported Centers in Some Selected Northern Nigeria States between 2019 - 2023.**

| | 2019 | 2020 | 2021 | 2022 | 2023 | Total (%) |
|---|---|---|---|---|---|---|
| Number Tested | | | | | | |
| FSW | 19,423 | 12,320 | 37,471 | 28,650 | 20,639 | 118,503 (36.5) |
| MSM | 10,468 | 6,640 | 34,804 | 21,833 | 15,507 | 89,254 (27.5) |
| PWID | 12,107 | 7,029 | 43,056 | 32,646 | 21,796 | 116,634 (36.0) |
| **Total** | **41,998** | **25,989** | **115,331** | **83,130** | **57,942** | **324,391 (100.0)** |
| Number Positive | | | | | | |
| FSW | 2,525 | 2,193 | 3,185 | 2,951 | 1,713 | 12,567 (41.1) |
| MSM | 827 | 757 | 1,601 | 1,834 | 1,070 | 6,089 (19.9) |
| PWID | 1,247 | 1,469 | 3,057 | 3,689 | 2,463 | 11,925 (39.0) |
| **Total** | **4,599** | **4,419** | **7,843** | **8,474** | **5,246** | **30,581 (100.0)** |
| Positivity Rate (%) | | | | | | |
| FSW | 13.0 | 17.8 | 8.5 | 10.3 | 8.3 | (10.6) |
| MSM | 7.9 | 11.4 | 4.6 | 8.4 | 6.9 | (6.8) |
| PWID | 10.3 | 20.9 | 7.1 | 11.3 | 11.3 | (10.2) |
| **Total** | **11.0** | **17.0** | **6.8** | **10.2** | **9.1** | **(9.4)** |
| Linkage Rate (%) | | | | | | |
| FSW | 100.2 | 108.3 | 103.9 | 98.0 | 103.0 | |
| MSM | 88.8 | 106.2 | 103.0 | 98.0 | 98.9 | |
| PWID | 92.5 | 116.3 | 93.6 | 99.6 | 100.0 | |
| Total | 96.0 | 110.6 | 99.7 | 98.7 | 100.7 | |
| Number Initiated on ART | | | | | | |
| FSW | 2,529 | 2,375 | 3,309 | 2,892 | 1,764 | 12,869 (41.8) |
| MSM | 734 | 804 | 1,649 | 1,798 | 1,058 | 6,043 (19.6) |
| PWID | 1,153 | 1,709 | 2,862 | 3,673 | 2,463 | 11,860(38.5) |
| **Total** | **4,416** | **4,888** | **7,820** | **8,363** | **5,285** | **30,772(100.0)** |
| Number Receiving Treatment (ART) | | | | | | |
| FSW | 2,387 | 4,703 | 7,918 | 10,218 | 12,769 | |
| MSM | 737 | 1,521 | 3,132 | 4,734 | 6,672 | |
| PWID | 1,064 | 2,750 | 5,585 | 8,844 | 11,472 | |
| **Total** | **4,188** | **8,974** | **16,635** | **23,796** | **30,913** | |
| Number With VL Sample Collected | | | | | | |
| FSW | 1,235 | 3,710 | 7,145 | 8,696 | 9,076 | |
| MSM | 371 | 1,193 | 2,768 | 3,885 | 4,506 | |
| PWID | 450 | 2,062 | 4,805 | 7,199 | 7,762 | |
| **Total** | **2,056** | **6,965** | **14,718** | **19,780** | **21,344** | |
| VL Testing Coverage (%) | | | | | | |
| FSW | 52 | 79 | 90 | 85 | 71 | |
| MSM | 50 | 78 | 88 | 82 | 68 | |
| PWID | 42 | 75 | 86 | 81 | 68 | |
| **Total** | **49** | **78** | **89** | **83** | **69** | |
| Number Virally Suppressed | | | | | | |
| FSW | 1,118 | 3,457 | 6,957 | 8,643 | 8,964 | |
| MSM | 330 | 1,103 | 2,691 | 3,872 | 4,424 | |
| PWID | 413 | 1,795 | 4,689 | 6,607 | 7,689 | |
| **Total** | **1,861** | **6,355** | **14,337** | **19,122** | **21,077** | |

*(Continued)*

**Table 1.** (Continued)

| | 2019 | 2020 | 2021 | 2022 | 2023 | Total (%) |
|---|---|---|---|---|---|---|
| VL Suppression Rate (%) | | | | | | |
| FSW | 91 | 93 | 97 | 99 | 99 | |
| MSM | 89 | 92 | 97 | 100 | 98 | |
| PWID | 92 | 87 | 98 | 92 | 99 | |
| **Total** | **91** | **91** | **97** | **97** | **99** | |

MSM – Men who have sex with men.; FSW – Female sex workers; PWID – People who inject drugs.; SFH – Society for Family Health, Nigeria

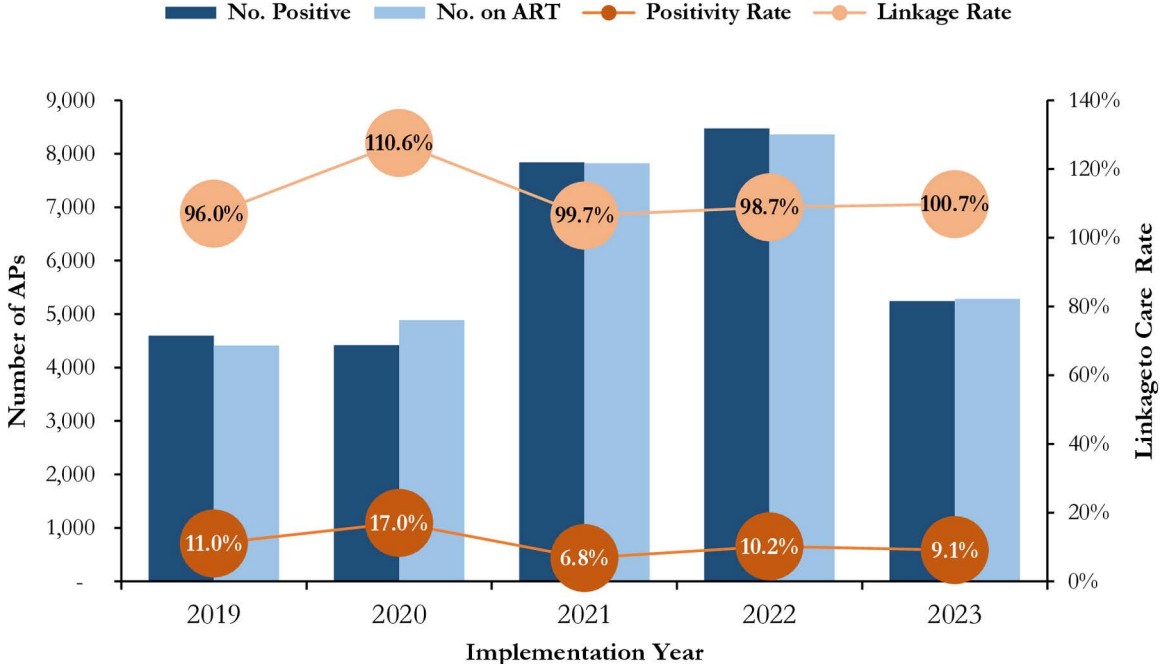

**Fig 4. Trends in HIV Case Finding, Positivity Rate and Linkage to Care among Affected Populations (MSM, FSW, PWID) Receiving HIV Care in SFH Supported Centers in Some Selected States in Northern Nigeria between 2019 - 2023.**

northern states where the SFH project is being implemented. Several studies in the Middle East, North Africa and other Muslim dominated countries in Asia such as Malysia have reported similar experiences in implementing AP programs due to stigma, harassment, homophobia, and criminalization thereby making the population to remain hidden [35–39]. Among the AP groups, MSM is the most hidden group [40,41] and therefore access to them requires careful understanding of their behaviors and innovations like the ones explained in this paper. This reason may likely explain why the number of MSM reached with testing is slightly lower than FSW and PWID. This indicates the need for AP service implementers to continue exploring more innovative strategies to reach MSMs to circumvent evolving new structural barriers. The number of HIV positives among PWID subpopulation has consistently doubled that of MSM from 2020 – 2023. The number of HIV positive PWID progressively increased from 2019 – 2022 indicating potential growth in the number of this subpopulation. Drug use among youths in northern Nigeria is a serious social problem that requires urgent intervention [42,43]. In the past, SFH pioneered harm reduction program

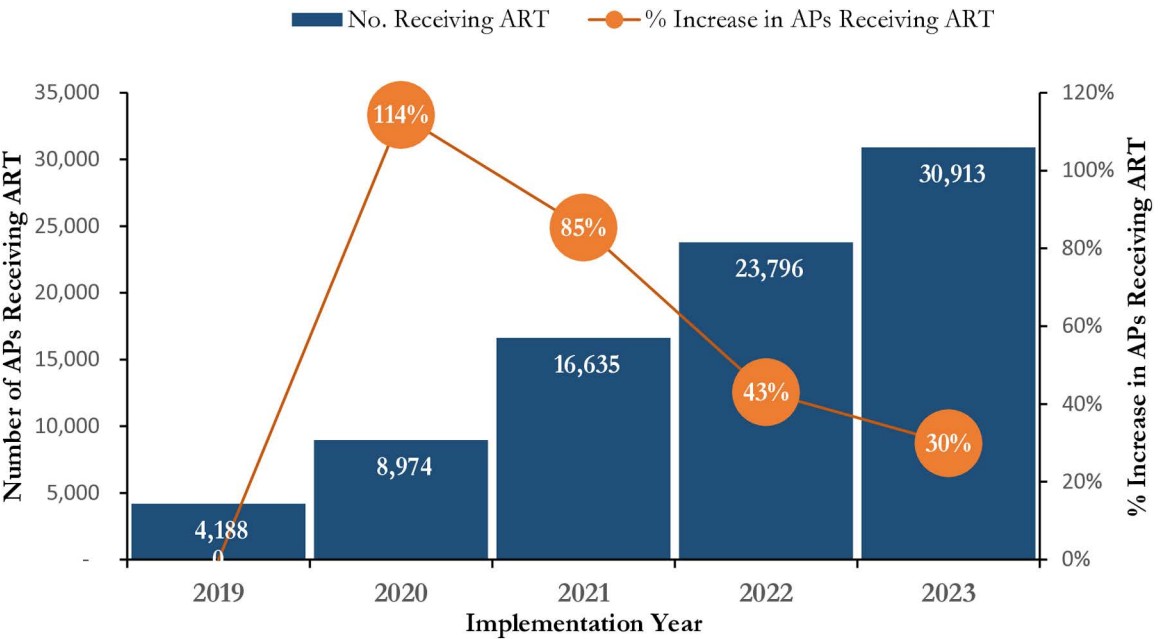

**Fig 5. Trends in Number of Affected Populations (MSM, FSW, PWID) Receiving HIV Care in SFH Supported Centers in Some Selected States in Northern Nigeria between 2019 - 2023.**

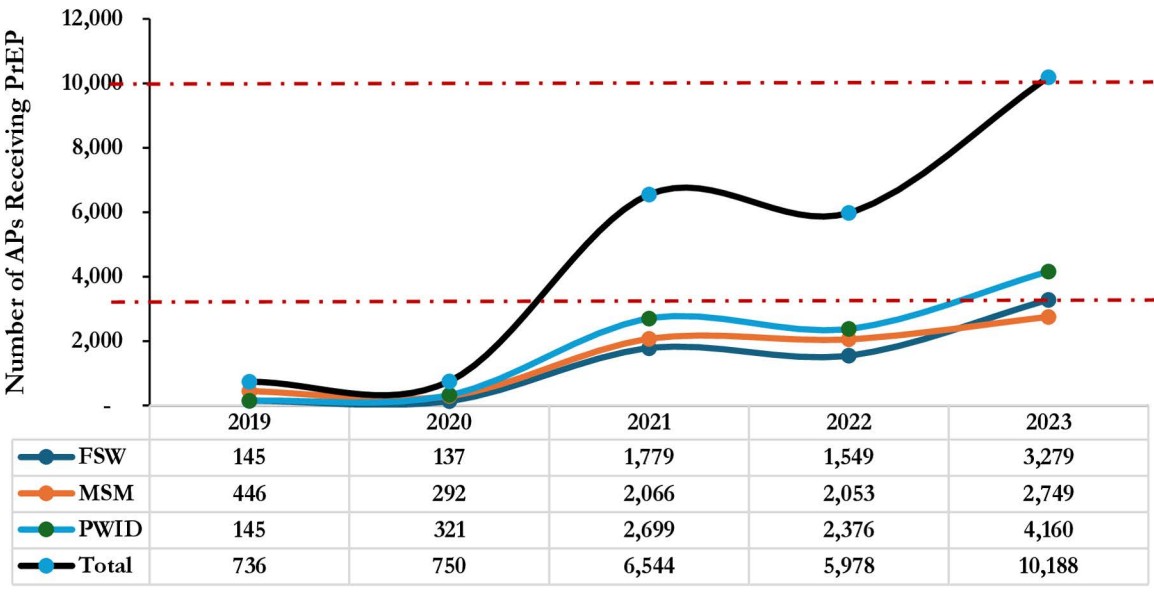

| | 2019 | 2020 | 2021 | 2022 | 2023 |
|---|---|---|---|---|---|
| FSW | 145 | 137 | 1,779 | 1,549 | 3,279 |
| MSM | 446 | 292 | 2,066 | 2,053 | 2,749 |
| PWID | 145 | 321 | 2,699 | 2,376 | 4,160 |
| Total | 736 | 750 | 6,544 | 5,978 | 10,188 |

**Fig 6. Trends in PrEP Uptake among Affected Populations (MSM, FSW, PWID) Receiving HIV Care in SFH Supported Centers in Some Selected States in Northern Nigeria between 2019 - 2023.**

in Nigeria where syringes and needles were given free to PWIDs in addition to social and behavioral change. Presently, there are several programs and rehab centers focusing on drug use among youths [44], however, most of these programs do not provide free syringes and needles or harm reduction interventions. Mostly the programs are limited to behavioral change and awareness creation. Therefore, this finding indicates the need to review these

programs by incorporating interventions that would minimize HIV spread among the subpopulations.

Linkage to care rate was maintained at the range of 96% – 110% over the 5-year period. The universal test and treat policy launched by Nigerian authorities in 2016 certainly contributed to this success. Equally central to this growth in linkage to care was active linkage to care for participants who declined to initiate ART immediately either for personal reasons or contraindications such as presumptive TB cases. Active linkage to care is shown to improve treatment initiation rate when clients are identified in the communities [45]. The success of our active linkage to care strategy is responsible for the spike of 110.6% linkage rate seen in the year 2020 and of 100.7% in 2023. Clients with presumptive TB cases were ineligible for universal test and treat because of the need for confirmatory TB test which usually takes between couple of days to up to a week. To ensure that presumptive TB cases who are diagnosed with HIV equitably commenced early ART, there is a need to devise an efficient approach to aid seamless TB confirmatory test to minimize leakages among HIV positive presumptive TB cases. The impact of the sustained linkage rate coupled with efficient testing strategy deployed over the 5-year period led to a 7-fold increase in the number of AP receiving ART from about 4,188 to about 30,000. Our client centered DSD models when deployed in year 1 (2019) started to yield results toward the end of the year with exponential growth between 2020 and 2021. Aspiration to attain unprecedented annual targets also motivated the team to put extraordinary efforts "the Surge" which involved intensive case finding, sustained linkage to care, optimal continuity of treatment, and effective return to care campaigns for clients that were lost to follow-up. The Surge also contributed to the exponential trajectory observed between 2019 – 2021. While the strategies explained in this article have proven to be effective in improving program performance, attaining annual program targets may not always be realized due to overambitious targets being set by funders, therefore, the Surge concept in addition, serves as strategy being deployed to accelerate programs to attain targets within shortest period when programs are lagging.

In Year 1, when VL sample collection strategies were not fully rolled out, VL testing coverage was as low as 42% among PWID and the highest was 52% among FSW with overall coverage of 49% among all the 3 AP typologies. However, with the introduction of strategies to address the prevailing challenges as described earlier such as flagging of AP due for VL test 4 weeks ahead of due date with constant follow-up by case managers and VL champions and other supportive measures such as home sample collection, reimbursement of transportation fares and aligning ARV refills with VL sample collection dates, there was steady progress with overall collection coverage of 86% in Year 3. Several studies shared similar experiences, for example by using dedicated support staff to support individuals eligible for VL test, VL testing coverage improved from 27% to 71% in urban health facility in Malawi [46]. Similarly, a study conducted to explore challenges associated with VL testing in resource limited settings suggested nonconventional facility-based sample collection as acceptable measures that can tremendously improve VL sample collection and coverage [47]. The observed decline in VL coverage between Year 4 – Year 5 was due to aggressive case finding conducted during the periods thereby overwhelming the available VL testing capacities of the PCR laboratories that led to delayed VL sample processing. However, this challenge has been addressed by the government in collaboration with donor agencies by improving the capacities of PCR laboratories to test more samples. Viral load (VL) suppression is maintained at above 90% throughout the 5-year period with progressive upward trajectory to 97% in 2021 – 2022 and to 99% in 2023. High VL suppression indicates program quality. Optimal linkage to care and timely ART initiation, adherence and good continuity of treatment, timely tracking and returning of IIT to care would all culminate in very good VL suppression [48,49] as seen in our program. A

study in the United States reported similar experiences whereby timely treatment initiation is associated with greater success in VL suppression while VL rebound was associated with irregular care [50]. Another study from Thailand argued that timely ART initiation is associated with greater chances of retention in care which certainly will improve VL suppression [51]. However, this study didn't consider universal test and treat policy whereby individuals initiate treatment and dropout early as reported from a study in Tanzania [52]. Contrary to the findings from Tanzania, in a time-series study from Uganda, it was found that universal test and treat policy significantly led to an immediate increase in retention rates though retention decreased for cohorts enrolled each month [53]. SFH's continuity of treatment enhancement strategies as discussed in this paper helped in circumventing the IIT challenges observed in Tanzania and Uganda thereby contributing to the successes seen in this study.

Pre-exposure prophylaxis-uptake leveled at about 750 per year between 2019 – 2020 and improvements were only recorded in 2021 following the introduction of SFH's PrEP uptake optimization initiatives. The initiatives include intensive client sensitization, education and awareness creation and introduction of PrEP champions. Similar findings were reported in Cameroon whereby improved uptake of PrEP was observed following sensitization and awareness creation among AP [54]. Expanding access to PrEP using initiatives such as pharmacy-based PrEP and telemedicine-delivered PrEP have been found to tremendously improve uptake of PrEP among AP in southern United States [55]. This finding is in line with SFH's DSD models approaches whereby services were taken to AP doorsteps. The southern US study also acknowledged the impact of addressing structural barriers in improving uptake of PrEP. Addressing structural barriers in AP program implementation is critical strategy in SFH's service delivery approaches and we believe has equally contributed to the PrEP uptake in our program. Findings from Portugal highlighted the need for continuous sensitization and education of AP in using PrEP. Only 10.9% of respondents from the Portugal's study reported some knowledge about PrEP with as low as 0.4% acknowledging using it [56]. We observed exponential growth in the uptake of PrEP after initiation continuous sensitization and education of AP in OSS and other DSD outlets in our program which also strengthened the findings from Portugal's study. Therefore, AP program implementers should pay attention to this simple but highly impactful strategy when trying to improve the uptake of new initiatives such as the PrEP.

This study is subject to a major limitation that it was not from inception designed to assess the impact of the strategies discussed in this paper. There may likely be some confounders that would have influenced some of the findings here whether positively or negatively. However, it is our hope that the strategies discussed would serve as good lessons for other AP implementing programs in similar settings toward minimizing barriers and ensuring equitable access to quality HIV care by AP.

## Conclusion and recommendations

The implementation of people-centered, evidence-driven, culturally, and religiously sensitive program has enabled SFH to reach a significantly high number of AP with the services they need in northern Nigeria. This helps improved equity in access to AP generally, but especially AP in northern Nigeria, despite the prevailing stigma and discrimination against AP in the region. There are specific program areas that need continuous improvement including strategies to reach MSMs to avoid the evolution of new structural barriers; expansion of PWID programming to optimize all aspects of harm reduction; and sustained sensitization, education, and awareness creation among AP to improve uptake of PrEP and other prevention and care services.

## Supporting information

**S1 Raw Data:  Datasets used in the study.**
(XLSX)

## Acknowledgements

We acknowledge the support of the United States Agency for International Development (USAID) in advancing the mission of the AP Care 2 project to ensure equitable access to quality HIV care for key populations in Nigeria. Similarly, we are grateful to the Foreign Commonwealth & Development Office (FCDO), formerly the Department for International Development (DFID) and the Global Fund (GF) for their support. We further acknowledge the support of the ministry of health in the study states for the supports and the enabling environment to support AP in their states.

## Author contributions

**Conceptualization:** Abdulsamad Salihu, Ibrahim Jahun, David Olusegun Oyedeji, Jennifer Anyanti.

**Data curation:** Abdulsamad Salihu, David Olusegun Oyedeji, Wole Fajemisin, Omokhudu Idogho, Samira Shehu.

**Formal analysis:** Abdulsamad Salihu, Ibrahim Jahun, Jennifer Anyanti.

**Funding acquisition:** Jennifer Anyanti.

**Investigation:** Abdulsamad Salihu, Jennifer Anyanti.

**Methodology:** Abdulsamad Salihu, Ibrahim Jahun, David Olusegun Oyedeji, Wole Fajemisin, Omokhudu Idogho, Samira Shehu, Jennifer Anyanti.

**Project administration:** Abdulsamad Salihu, Jennifer Anyanti.

**Resources:** Jennifer Anyanti.

**Supervision:** Jennifer Anyanti.

**Validation:** Abdulsamad Salihu, Ibrahim Jahun, David Olusegun Oyedeji, Jennifer Anyanti.

**Visualization:** Abdulsamad Salihu, Ibrahim Jahun.

**Writing – original draft:** Abdulsamad Salihu, Ibrahim Jahun.

**Writing – review & editing:** Abdulsamad Salihu, Ibrahim Jahun, David Olusegun Oyedeji, Wole Fajemisin, Omokhudu Idogho, Samira Shehu, Jennifer Anyanti.

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
