## [Decision Letter · Decision Letter 0]

2 Jan 2025

Dear Dr. Jahun,

Thank you for submitting your manuscript to PLOS ONE. After careful consideration, we feel that it has merit but does not fully meet PLOS ONE’s publication criteria as it currently stands. Therefore, we invite you to submit a revised version of the manuscript that addresses the points raised during the review process.

We look forward to receiving your revised manuscript.

Kind regards,

Okikiolu Badejo, M.D., Ph.D

Academic Editor

PLOS ONE

**Journal Requirements:**

This study has been supported by the President’s Emergency Plan for AIDS Relief (PEPFAR) through the United State Agency for International Development (USAID) under the terms of USAID Cooperative Agreement No. 7206201900009. The findings and conclusions in this report are those of the authors and do not necessarily represent the official position of the funding agencies.

The authors acknowledge the enormous contributions from the United States Agency for International Development (USAID). The support and contributions from USAID have been instrumental in advancing the mission of the KP CARE 2 Project to ensure equitable access to quality HIV care for key populations in Nigeria. The Society for Family Health (SFH) would like to express its gratitude for the financial support received, which has enabled us to address the complex socio-cultural and structural barriers faced by key populations in Nigeria.

We also acknowledge the significant support from the Foreign, Commonwealth & Development Office (FCDO), formerly the Department for International Development (DFID), and the Global Fund. Their funding and partnership have been vital in supporting the Society for Family Health (SFH) over the years, enabling us to implement various HIV support projects and improve health outcomes across Nigeria. These collaborative efforts underscore the critical role of international partnerships in addressing global health challenges and achieving impactful, sustainable outcomes.

This study has been supported by the President’s Emergency Plan for AIDS Relief (PEPFAR) through the United State Agency for International Development (USAID) under the terms of USAID Cooperative Agreement No. 7206201900009. The findings and conclusions in this report are those of the authors and do not necessarily represent the official position of the funding agencies.

Reviewers' comments:

Reviewer's Responses to Questions

**Comments to the Author**

1. Is the manuscript technically sound, and do the data support the conclusions?

Reviewer #1: Partly

Reviewer #2: Partly

2. Has the statistical analysis been performed appropriately and rigorously?

Reviewer #1: No

Reviewer #2: N/A

3. Have the authors made all data underlying the findings in their manuscript fully available?

Reviewer #1: Yes

Reviewer #2: Yes

4. Is the manuscript presented in an intelligible fashion and written in standard English?

Reviewer #1: Yes

Reviewer #2: Yes

**Reviewer #1: ** There are a lot of gaps in the methods section. The authors have not made it clear how the analysis was conducted; they make mention of the data being publicly sourced but do not clearly state the datasets that were used for this analysis, the software used for the analysis, and the time period in which this data was sourced.

I think the methods section should be re-worked to demonstrate the rigor and reproducibility of the results they have written up

**Reviewer #2:**  This paper provided an account of programme strategies to increase access to HIV prevention (PrEP), testing and treatment services for members of key populations living with HIV (KPLHIV) in northern Nigeria, a context of legal and sociocultural barriers that impede service delivery. This study has the potential to provide evidence and guidance to policymakers, implementers, and practitioners on how to adopt or adapt best practices to meet the needs of KPLHIV.

General comment

• This paper is regarded as original research, but the study design, data collection, and analysis are not explained in detail. While the authors mentioned the source of data for the secondary data, the type of data collected (variables) and operational definition of key indicators/outcomes (linkage to care rate) are lacking. Authors should include this information for clarity.

• When introducing acronyms, authors should ensure they are maintained throughout the document for consistency. For example, KPs and KP are used interchangeably to mean Key populations.

• Authors should ensure references (missing citations) are included for data reported and facts.

• Nigeria is a big country with many regions and states. To better understand the context and situate findings, authors should state the specific northern regions (e.g. Northcentral, Northwestern, etc) and states where SFH implemented these novel strategies in Nigeria

• What is the estimated population of KP in northern Nigeria (from KP size estimates, if any) to understand the treatment coverage gap and the extent to which the programme/strategies are closing the gap

• It is not clear whether HIV services were offered to KP in or through the same location or model of care for all KP groups or specific subgroups (FSW or MSM). Are these strategies implemented in the same manner for all KP groups, or are they different per KP group?

• This manuscript is silent on the sub-optimal coverage of viral load testing in the programme. This information is crucial because it reflects on the resources available in the program and access to essential HIV/AIDS services. Contextualize your finding of sub-optimal VL coverage in northern Nigeria despite a seemingly good retention rate (as demonstrated by the increasing number of clients on ART in your programme)

• Expound on the VL testing trend, barriers to accessing VL testing in the programme, and solutions in the discussion section

Specific comments

Abstract (results)

- Access to viral load testing services is crucial to optimal treatment outcomes. Authors should consider including performance or trend in viral load coverage in their results.

Main Article

Introduction

• Paragraph 1, lines 2-3 reads: ‘’…..about 39 million people (about twice the population of New York)’’. Since this study is situated in sub-Saharan Africa, perhaps relating this population to a region, country or city in Africa would be more meaningful to readers.

• Paragraph 2, lines 2-4 states: ‘’KPs are defined groups who…..’’ Authors should provide a reference(s) for this definition.

• Provide a reference for the second statement in paragraph two

• Edit decriminalize to criminalize in this statement: ''……for example, Nigeria is among the country that decriminalizes lesbians, gay, bisexual ….(LGBTQI).''

Methods

• There is no mention of the specific study design used for this study. Secondary data analysis is not a study design.

• Information about data sources and analysis is coming a bit early. I suggest a subheading titled "data collection and analysis."

• The description of the study area should include the names of states in northern Nigeria where SFH is implementing the KP programme.

o Refer to KP size estimates in Nigeria (if any) to better understand the context and the extent to which your KP program strategies are closing the treatment gaps.

• While the authors mentioned the source of data for the secondary data, they did not describe the type of data collected (variables) and the operational definition of key indicators/outcomes (linkage to care rate). Authors should include this information for clarity.

Strategies

• Authors should consider rephrasing the subheadings for consistency and clarity: rephrase the second strategy to align with the first one. ……’’addressing systems barriers vs addressing structural barriers’’.

• What is retainer-ship? elaborate on this for clarity

Data management

• Authors often used adjectives to qualify their project input and activities or processes, such as robust data management practices, high-quality data, and highly skilled data management staff. Although authors operationalize these qualities by describing their unique characteristics or elements, they can be modest in using superfluous adjectives while describing their research.

Result

• The authors showed project performance along the cascade of HIV care and trends in HIV case finding, linkage to ART, current on ART, and PrEP uptake. In the method section, authors should describe the data considered and variables collected. Also, definitions for indicators such as HIV positivity rate, linkage to care, and viral suppression rate should be provided.

• The overall viral load coverage is sub-optimal, 69%. Add a comment on this in the results section and discuss the VL coverage trend, barriers to accessing VL testing in the programme, and solutions in the discussion section

• While the number of clients accessing PrEP increased over the study period, authors should consider presenting data on PrEP uptake/coverage among eligible clients (PrEP offer). This is more informative than the number of clients accessing PrEP services in the program.

Discussion

• Paragraph 1: reference this statement: ‘’….Presently, several programs focus on drug use…………………harm reduction interventions''.

• For clarity, rephrase this statement: ‘’The …..indicating a potential surge in this sub-population’’. What do authors mean by ‘’surge’’?

• How does the ''surge intervention'' differ from the strategies described in the method section? Expatiate on this

• Many studies/programmes have described sub-optimal viral load coverage in KP programmes but a reasonable viral load suppression rate. This information is essential because it reflects on the resources available in the program and access to critical HIV services. Contextualize your finding of sub-optimal VL coverage in northern Nigeria despite a seemingly good retention rate (as demonstrated by the increasing number of clients on ART in your programme)

o Discuss the VL trend, barriers to accessing VL testing in the programme, and solutions in the discussion section

• In paragraph 4, the authors write: ‘’…..findings from Portugal buttressed the need for continuous sensitization and education of KPs…..just like we reported in this study’’. Kindly rephrase this statement, as authors did not assess or report the effect of KP sensitization and education on HIV services uptake in this paper/study

• Last paragraph: typographical error----compounders vs confounders

Figure 1:

• Introduce legends or labels to show or differentiate strategies from activities

• DSD*= edit to remove or define the asterisk

**Do you want your identity to be public for this peer review?** For information about this choice, including consent withdrawal, please see our Privacy Policy

Reviewer #1: No

Reviewer #2: No

---

## [Author Response · Author response to Decision Letter 1]

20 Jan 2025

Comment Response

Academic Editor

● Please provide an amended statement that declares *all* the funding or sources of support (whether external or internal to your organization) received during this study.

Please also include the statement “There was no additional external funding received for this study.” in your updated Funding Statement. Thanks for the comment. Please find below the edited funding statement.

“This study was supported by the President’s Emergency Plan for AIDS Relief (PEPFAR) through the United State Agency for International Development (USAID) under the terms of USAIDS Cooperative Agreement No. 7206201900009. There was no additional external funding received for this study.

The findings and conclusions in this report are those of the authors and do not necessarily represent the official position of the funding agencies.”

● We note that you have provided funding information that is not currently declared in your Funding Statement. However, funding information should not appear in the Acknowledgments section or other areas of your manuscript. We will only publish funding information present in the Funding Statement section of the online submission form.

Please remove any funding-related text from the manuscript and let us know how you would like to update your Funding Statement. Thanks for the comment. The acknowledgement section has been updated as follows:

“We acknowledge the support of the United States Agency for International Development (USAID) in advancing the mission of the KP Care 2 project to ensure equitable access to quality HIV care for key populations in Nigeria. We further acknowledge the support of the ministries of health in the study states which created the enabling environment to provide KPs with the care and services they need”

● Please include captions for your Supporting Information files at the end of your manuscript, and update any in-text citations to match accordingly Thanks for the comments. The following caption for supporting document has been added at the end of the manuscript:

S1 Raw Data (Datasets used in the study).

The file is labelled as:

S1_Raw Data

Reviewer 1

● There are a lot of gaps in the methods section. The authors have not made it clear how the analysis was conducted; they make mention of the data being publicly sourced but do not clearly state the datasets that were used for this analysis, the software used for the analysis, and the time period in which this data was sourced. Thanks for the comment. The methods section has been updated accordingly. We have updated subsection “study design and data access”, added subsection “data collection and analysis” and provided operational definitions of all variables and indicators used in the study.

Additionally, details about period when data was sourced and analysed, software used (MS Excel) have now been included under subsection “data collection and analysis”

Reviewer 2

General comments

● This paper is regarded as original research, but the study design, data collection, and analysis are not explained in detail. While the authors mentioned the source of data for the secondary data, the type of data collected (variables) and operational definition of key indicators/outcomes (linkage to care rate) are lacking. Authors should include this information for clarity.

Thank you. The study design is now added and the subsection “study design and data access” has been moved to the end of the “methods” section.

Additionally, we created a subsection “data collection and analysis” whereby we concisely provided steps about how data were collected, reviewed and analysed. We have also provided the operational definitions of variables, indicators and outcomes accordingly.

● When introducing acronyms, authors should ensure they are maintained throughout the document for consistency. For example, KPs and KP are used interchangeably to mean Key populations.

Thanks for the comments. Acronym KP is now maintained throughout the manuscript.

● Authors should ensure references (missing citations) are included for data reported and facts.

Thanks for the comments. All missing references have been added.

● Nigeria is a big country with many regions and states. To better understand the context and situate findings, authors should state the specific northern regions (e.g. Northcentral, Northwestern, etc) and states where SFH implemented these novel strategies in Nigeria

Thank you for the observation. The following statement has been added at the beginning of subsection “strategies”:

“The strategies described in this manuscript were implemented in all states where SFH provides KPs services, and they were tailored to suit the needs of all KPs typologies. However, the data used in this study were obtained from six northern states (3 states each from northwest and northeast) as illustrated in Figure 1.”

Additionally, Figure 1 is added to illustrated geographical locations of the states where SFH has implemented these strategies and the 6 northern study states which the data used for the study was obtained from.

● What is the estimated population of KP in northern Nigeria (from KP size estimates, if any) to understand the treatment coverage gap and the extent to which the programme/strategies are closing the gap

Thank you for the suggestion. Unfortunately, the available KP size estimates in Nigeria were not done for all the states in the country. Of the 6 northern states used in this study, KP size estimates are available for only 1 state. We feel it may be confusing to readers to include this context since there is no data for the other 5 study states.

● It is not clear whether HIV services were offered to KP in or through the same location or model of care for all KP groups or specific subgroups (FSW or MSM). Are these strategies implemented in the same manner for all KP groups, or are they different per KP group?

Thank you for the observation. This has now been clearly articulated in the “methods” section in the first paragraph of subsection “strategies” as follows:

“The strategies described below were implemented in all states where SFH provides KPs services, and they were tailored to suit the needs of all KPs typologies”.

● This manuscript is silent on the sub-optimal coverage of viral load testing in the programme. This information is crucial because it reflects on the resources available in the program and access to essential HIV/AIDS services. Contextualize your finding of sub-optimal VL coverage in northern Nigeria despite a seemingly good retention rate (as demonstrated by the increasing number of clients on ART in your programme)

● Expound on the VL testing trend, barriers to accessing VL testing in the programme, and solutions in the discussion section

Thank you for the comment. We have updated Table 1 by adding “VL Testing Coverage”. Similarly, the results section narrative has been updated as follows:

“There was progressive increase in VL testing coverage between Year 1 – Year 3 across all the three KPs typologies and then steady decline between Year 4 – Year 5.”

Additionally, “discussion section” has been updated as follows:

“In Year 1, when VL sample collection strategies were not fully rolled out, VL testing coverage was as low as 42% among PWID and the highest was 52% among FSW with overall coverage of 49% among all the 3 KPs typologies. However, with the introduction of strategies to address the prevailing challenges as described earlier such as flagging of KPs due for VL test 4 weeks ahead of due date with constant follow-up by case managers and VL champions and other supportive measures such as home sample collection, reimbursement of transportation fares and aligning ARV refills with VL sample collection dates, there was steady progress with overall collection coverage of 86% in Year 3. Several studies shared similar experiences, for example by using dedicated support staff to support individuals eligible for VL test, VL testing coverage improved from 27% to 71% in urban health facility in Malawi [48]. Similarly, a study conducted to explore challenges associated with VL testing in resource limited settings suggested nonconventional facility-based sample collection as acceptable measures that can tremendously improve VL sample collection and coverage [49]. The observed decline in VL coverage between Year 4 – Year 5 was due to aggressive case finding conducted during the periods thereby overwhelming the available VL testing capacities of the PCR laboratories that led to delayed VL sample processing. However, this challenge has been addressed by the government in collaboration with donor agencies by improving the capacities of PCR laboratories to test more samples”

Specific comments

Abstract (results)

- Access to viral load testing services is crucial to optimal treatment outcomes. Authors should consider including performance or trend in viral load coverage in their results.

Thank you for the comment. This has been addressed in the abstract and in the main article under result. Additionally, Table 1 has been updated with VL testing coverage results.

Main Article

Introduction

● Paragraph 1, lines 2-3 reads: …. about 39 million people (about twice the population of New York)’’. Since this study is situated in sub-Saharan Africa, perhaps relating this population to a region, country or city in Africa would be more meaningful to readers.

Thanks for the suggestion. This has been edited to read as “.... about 2.5 times the population of Lagos”.

● Paragraph 2, lines 2-4 states: ‘’KPs are defined groups who….’’ Authors should provide a reference(s) for this definition.

Thank you for the comment. Reference inserted [4] Pan American Health Organization (PAHO): Key populations. Available from: https://www.paho.org/en/topics/key-populations#

● Edit decriminalize to criminalize in this statement: ''……for example, Nigeria is among the country that decriminalizes lesbians, gay, bisexual …. (LGBTQI).''

Thank you for the comment. This has been edited accordingly (decriminalize edited to criminalize).

Methods

● There is no mention of the specific study design used for this study. Secondary data analysis is not a study design Thanks for the comment. Study design has been updated under subsection Study design and data access as follows: “This was retrospective cross-sectional study in a form of secondary data analysis using deidentified aggregate routine program data from Jan 2019 – December 2023”

● Information about data sources and analysis is coming a bit early. I suggest a subheading titled "data collection and analysis."

Thank you for the suggestion. A subheading “data collection and analysis” has been created after data management at the end of “Methods” section.

● The description of the study area should include the names of states in northern Nigeria where SFH is implementing the KP programme.

o Refer to KP size estimates in Nigeria (if any) to better understand the context and the extent to which your KP program strategies are closing the treatment gaps.

Thank you for the comment. We have created a map of Nigeria (Figure 1) with 2 categories of states: 1 - All states in Nigeria where SFH is implementing the KP programme, and 2 - States with data included in this study (the study states)

Thanks so much for the insight and great suggestion. This is much appreciated. Unfortunately, the available KP size estimates in Nigeria were not done for all the states in the country. Of the 6 northern states used in this study, KP size estimates were available for only 1 state. We feel it may be confusing for readers to include this context since there is no data for the other 5 study states.

● While the authors mentioned the source of data for the secondary data, they did not describe the type of data collected (variables) and the operational definition of key indicators/outcomes (linkage to care rate). Authors should include this information for clarity.

Thank you. We have now included operational definitions of all variables analysed in this study under “methods section”, subsection “data collection and analysis”.

● Authors should consider rephrasing the subheadings for consistency and clarity: rephrase the second strategy to align with the first one. ……’’addressing systems barriers vs addressing structural barriers’’.

Thanks for comments. “Systems barriers (service-provider and client-related barriers” has been edited as “Addressing systems barriers (service-provider and client-related barriers”.

● What is retainer-ship? elaborate on this for clarity

● Thanks for the comment. We have elaborated about “retainership” as follows:

“... an arrangement in which SFH engages services of lawyers to serve affected KPs on a recurring basis in exchange for a regular fee (retainer fee)”

Data management

● Authors often used adjectives to qualify their project input and activities or processes, such as robust data management practices, high-quality data, and highly skilled data management staff. Although authors operationalize these qualities by describing their unique characteristics or elements, they can be modest in using superfluous adjectives while describing their research.

Thanks for the observations. We have adjusted this accordingly throughout the manuscript.

● The authors showed project performance along the cascade of HIV care and trends in HIV case finding, linkage to ART, current on ART, and PrEP uptake. In the method section, authors should describe the data considered and variables collected. Also, definitions for indicators such as HIV positivity rate, linkage to care, and viral suppression rate should be provided.

Thanks for this comment. We have now included operational definitions of all variables and indicators analysed in this study under “methods section”, subsection “data collection and analysis”.

● The overall viral load coverage is sub-optimal, 69%. Add a comment on this in the results section and discuss the VL coverage trend, barriers to accessing VL testing in the programme, and solutions in the discussion section

● Thank you for the comment. We have updated Table 1 by adding “VL Testing Coverage”. Similarly, the results section narrative has been updated as follows:

“There was progressive increase in VL testing coverage between Year 1 – Year 3 across all the three KPs typologies and then steady decline between Year 4 – Year 5.”

Additionally, “discussion section” has been updated as follows:

“In Year 1, when VL sample collection strategies were not fully rolled out, VL testing coverage was as low as 42% among PWID and the highest was 52% among FSW with overall coverage of 49% among all the 3 KPs typologies. However, with the introduction of strategies to address the prevailing challenges as described earlier such as flagging of KPs due for VL test 4 weeks ahead of due date with constant follow-up by case managers and VL champions and other supportive measures such as home sample collection, reimbursement of transportation fares and aligning ARV refills with VL sample collection dates, there was steady progress with overall collection coverage of 86% in Year 3. Several studies shared similar experiences, for example by using dedicated support staff to support individuals eligible for VL test, VL testing coverage improved from 27% to 71% in urban health facility in Malawi [48]. Similarly, a study conducted to explore challenges associated with VL testing in resource limited settings suggested nonconventional facility-based sample collection as acceptable measures that can tremendously improve VL sample collection and coverage [49]. The observed decline in VL coverage between Year 4 – Year 5 was due to aggressive case finding conducted during the periods thereby overwhelming the available VL testing capacities of the PCR laboratories that led to delayed VL sample processing. However, this challenge has been addressed by the government in collaboration with donor agencies by improving the capacities of PCR laboratories to test more samples”

● While the number of clients accessing PrEP increased over the study period, authors should consider presenting data on PrEP uptake/coverage among eligible clients (PrEP offer). Thi

---

## [Decision Letter · Decision Letter 1]

9 Feb 2025

Ensuring Equitable Access to Quality HIV Care for Key Populations in Complex Sociocultural Settings: Lessons from Nigeria.

PONE-D-24-30895R1

Dear Dr. Jahun,

We’re pleased to inform you that your manuscript has been judged scientifically suitable for publication and will be formally accepted for publication once it meets all outstanding technical requirements.

Kind regards,

Okikiolu Badejo, M.D., Ph.D

Academic Editor

PLOS ONE

Additional Editor Comments (optional):

Reviewers' comments:

Reviewer's Responses to Questions

**Comments to the Author**

Reviewer #1: (No Response)

Reviewer #2: All comments have been addressed

2. Is the manuscript technically sound, and do the data support the conclusions?

Reviewer #1: Yes

Reviewer #2: Yes

3. Has the statistical analysis been performed appropriately and rigorously?

Reviewer #1: Yes

Reviewer #2: N/A

4. Have the authors made all data underlying the findings in their manuscript fully available?

Reviewer #1: Yes

Reviewer #2: Yes

5. Is the manuscript presented in an intelligible fashion and written in standard English?

Reviewer #1: Yes

Reviewer #2: Yes

Reviewer #1: NIL

Reviewer #2: (No Response)

**Do you want your identity to be public for this peer review?** For information about this choice, including consent withdrawal, please see our Privacy Policy

Reviewer #1: No

Reviewer #2: No

---

## [Editor Report · Acceptance letter]

PONE-D-24-30895R1

PLOS ONE

Dear Dr. Jahun,

I'm pleased to inform you that your manuscript has been deemed suitable for publication in PLOS ONE. Congratulations! Your manuscript is now being handed over to our production team.

Kind regards,

on behalf of

Dr. Okikiolu Badejo

Academic Editor

PLOS ONE